# Serum Alkaline Phosphatase as a Predictor of Cardiac and Cerebrovascular Complications after Lumbar Spinal Fusion Surgery in Elderly: A Retrospective Study

**DOI:** 10.3390/jcm8081111

**Published:** 2019-07-26

**Authors:** Ann Hee You, Dong Woo Han, Sung Yeon Ham, Wonsik Lim, Young Song

**Affiliations:** 1Department of Anesthesiology and Pain Medicine, Kyung Hee University Hospital, Seoul 02447, Korea; 2Department of Anesthesiology and Pain Medicine, Anesthesia and Pain Research Institute, Gangnam Severance Hospital, Yonsei University College of Medicine, Seoul 06237, Korea

**Keywords:** alkaline phosphatase, elderly, postoperative complication

## Abstract

We retrospectively enrolled 1395 patients aged > 65 years undergoing posterior lumbar spinal fusion surgery and classified them into tertiles based on serum Alkaline Phosphatase (ALP) levels (<63, 63–79, >79 IU/L). The primary outcome was the incidence of 30-day major adverse cardiac and cerebrovascular events (MACCE; composite endpoint defined as the occurrence of ≥1 of the following events: new-onset myocardial infarction, stroke, or cardiovascular mortality). The incidence of the composite endpoint was the highest in the third serum ALP tertile (0.4% vs. 0.2% vs. 2.2% in the first, second, and third tertile, respectively, *p* = 0.003). Multivariate analysis showed that the third serum ALP tertile was an independent predictor of the composite endpoint of MACCE (odds ratio 4.507, 95% confidence interval 1.378–14.739, *p* = 0.013). The optimal cut-off value of preoperative serum ALP showing the best discriminatory capacity to predict postoperative MACCE (measured by receiver-operating characteristic curve analysis) was 83 IU/L (area under curve 0.694, 95% confidence interval 0.574–0.813, *p* = 0.016). Preoperative serum ALP levels were independently associated with the composite endpoint of postoperative 30-days MACCE. We suggest that serum ALP can be used as a biomarker to predict cardiac and cerebrovascular complications following lumbar spinal fusion surgery in elderly patients.

## 1. Introduction

Owing to the global phenomenon of a rapidly aging population, greater longevity, and more active lifestyles in this patient population, the number of elderly patients undergoing surgery has increased rapidly over the past few decades [1]. Despite the lower and decreasing incidence of critical cardiovascular complications associated with non-cardiac surgery, age-related comorbidities remain a concerning issue [2,3]. Vascular calcification might be the most important contributor to perioperative cardiovascular events in the elderly patients, but having been concerned less compared to other cardiopulmonary illness [4,5]. Vascular imaging or functional studies are time-consuming, expensive, require a trained examiner, and are associated with the risk of adverse reactions to contrast dye. Therefore, simple molecular markers of vascular calcification may be an attractive option at perioperative period.

Vascular calcification is characterised by an imbalance between promoters and inhibitors of mineralisation induced by injury, renal insufficiency, or aging [6]. Inorganic pyrophosphate is a potent endogenous inhibitor of tissue mineralisation that inhibits the growth and tissue deposition of hydroxyapatite crystals [7]. Alkaline phosphatase (ALP) promotes hydrolysis inactivation of pyrophosphate and plays an important role in cardiovascular calcification observed in patients with uremia [6,8]. Reportedly, ALP activity was shown to be associated with vascular inflammation and endothelial dysfunction in patients with chronic kidney disease (CKD) [7]. Serum ALP level, a biomarker of liver or bone disease, has become to serve as an independent predictor of cardiovascular prognosis in patients with renal insufficiency [9]. Recent studies have reported the potential predictive power of serum ALP for overall prognosis not only in patients with stroke and myocardial infarction (MI), but even in the general population [10,11]. Moreover, in patients with coronary artery disease (CAD) undergoing stent implantation, higher baseline serum ALP levels effectively predicted cardiac or cerebrovascular events within 1 year following intervention [12].

Recently, an 11-year follow-up prospective study involving 3380 elderly men reported a significant association between higher serum ALP levels and the risk of CAD, MI, stroke, or mortality [13]. However, to date, no study has investigated whether serum ALP levels are associated with the risk of perioperative cardiovascular complications following non-cardiac surgery in elderly patients. We investigated the association between preoperative serum ALP levels and cardiac and cerebrovascular complications in elderly patients undergoing lumbar spinal fusion surgery.

## 2. Methods

### 2.1. Study Population

The protocol of the study was consistent with the ethical guidelines of the 2008 Helsinki Declaration. Institutional Review Board of Yonsei University Gangnam Severance Hospital ensured appropriate ethical and bioethical conduct and approved this study (No. 3-2017-0384). Written informed consent was waived due to its retrospective nature.

We retrospectively reviewed the electronic medical records of patients aged >65 years who underwent posterior lumbar spinal fusion surgery between March 2011 and May 2017 at the Yonsei University College of Medicine Gangnam Severance Hospital (*n* = 2049). Exclusion criteria for the study were as follows: established liver disease defined as obstruction of the bile ducts, hepatitis, or moderate-to-severe fatty liver (*n* = 112), bone pathology (*n* = 38), cancer (*n* = 181), previously diagnosed moderate-to-severe valvulopathy (*n* = 51), cerebrovascular attacks (*n* = 79), MI (*n* = 41), arrhythmia-induced hemodynamic instability (*n* = 31), dialysis performed for kidney failure (*n* = 41), left ventricular ejection fraction <40% (*n* = 55), and lack of data regarding preoperative serum ALP levels (*n* = 25). We classified patients into 3 groups by tertiles based on preoperative serum ALP levels.

### 2.2. Study Endpoints

The primary outcome was the incidence of 30-day major adverse cardiac and cerebrovascular events (MACCE, composite endpoint defined as the occurrence of ≥1 of the following events: new-onset MI, stroke, or cardiovascular mortality) [14]. MI was defined as an increase in the peak serum troponin-T isoenzyme level that was >5-fold the upper limit of the reference range or a newly developed Q wave. Stroke was defined as new-onset neurological deterioration secondary to ischemic brain injury diagnosed by a neurologist.

### 2.3. Other Assessments

We recorded baseline patient characteristics including age, sex, body mass index, American Society of Anesthesiologists (ASA) physical status classification, preoperative cardiovascular medications, smoking habits, comorbidities (hypertension, diabetes mellitus, CAD, chronic obstructive pulmonary disease, CKD, and cerebrovascular accidents). Additionally, we recorded preoperative laboratory data including serum ALP, phosphorus, aspartate transaminase, alanine transaminase, albumin, and total bilirubin levels. We also recorded perioperative data including the number of fusion levels, operation time, intraoperative input and output, requirement of postoperative mechanical ventilation, intensive care unit (ICU) admission, and length of hospital stay.

### 2.4. Statistical Analysis

All statistical analyses were performed using the Statistical Package for Social Sciences (ver. 23.0 for Windows, SPSS Inc., Chicago, IL, USA), and *p*-values < 0.05 were considered statistically significant.

We used the Shapiro-Wilk test for normality, and continuous variables among the tertiles were compared using one-way analysis of variance with the Bonferroni post hoc for normally distributed data or Kruskal-Wallis test with the Dunn procedure for data that were not normally distributed. We compared continuous variables between patients with and without MACCE using the Student *t*-test for normally distributed data and the Mann-Whitney U test for data that were not normally distributed. We used the chi-squared or the Fisher exact test for categorical variables. Continuous variables were expressed as mean ± standard deviation or median (interquartile range), and categorical variables were expressed as numbers (%).

We performed logistic regression analysis to evaluate predictors of postoperative 30-day MACCE with known risk factors including age, CAD, CKD, and estimated intraoperative blood loss as well as the variables showing *p* < 0.2 between patients with and without MACCE. Variables showing *p* < 0.2 after univariate analysis were selected using a stepwise selection method and subjected to multivariate logistic regression analysis. Predictability was expressed using an odds ratio (OR) and 95% CI. We used receiver-operating characteristic (ROC) curve analysis to determine the optimal cut-off value of preoperative serum ALP showing the best discriminatory capacity to predict postoperative MACCE. The optimal cut-off value was defined as the serum ALP level showing the greatest sum of sensitivity and specificity.

## 3. Results

A total of 1395 patients were included and analyzed. Based on preoperative serum ALP levels, we classified 468 patients into the first serum ALP tertile (< 63 IU/L), 463 patients into the second serum ALP tertile (63–79 IU/L), and 464 patients into the third serum ALP tertile (> 79 IU/L). No significant intergroup differences were observed in age, sex, BMI, comorbidities, smoking habits, preoperative laboratory data including liver enzymes, and cardiovascular medication except in the percentage of patients using beta-blockers preoperatively (9.2% vs. 13.0% vs. 7.8%, respectively, *p* = 0.024) (Table 1).

No significant intergroup differences were observed in operation time, the number of fusion levels, fluid balance, and estimated intraoperative blood loss (Table 2). An intergroup comparison of 30-day postoperative outcomes showed no differences; however, we observed a trend showing higher incidence of MI and stroke in patients in the third tertile than in patients in the other tertiles (0.4% vs. 0.2% vs. 1.3% of MI, *p* = 0.095 and 0.2% vs. 0% vs. 0.9% for stroke, *p* = 0.073 in first, second, and third tertile, respectively). The incidence of the composite endpoint was significantly higher in patients in the third tertile than in patients in the other tertiles (0.4% vs. 0.2% vs. 2.2%, in first, second, and third tertile, respectively, *p* = 0.003).

Based on the occurrence of 30-day MACCE, we categorized patients into two groups as follows: MACCE (*n* = 13) and No MACCE (*n* = 1382) (Table 3). Following intergroup comparison, female sex, diabetes, and serum ALP tertiles met the predefined criterion of *p* < 0.2 and were therefore subjected to logistic regression analysis combined with known risk factors for cardiovascular complications including age, CAD, CKD, and estimated intraoperative blood loss (Table 4). Using univariate analysis, we observed that female sex, diabetes, and the third serum ALP tertile showed an intergroup difference (*p* < 0.2) and were subjected to multivariate analysis. Multivariate analysis showed that the third serum ALP tertile (odds ratio 4.507, 95% CI 1.378–14.739, *p* = 0.013) remained an independent predictor of the composite endpoint of MACCE.

ROC curve analysis showed that the preoperative serum ALP level with the best discriminatory capacity to predict postoperative MACCE demonstrated an area under the curve of 0.694 with a *p*-value = 0.016 and 95% CI 0.574–0.813. The optimal cut-off value of preoperative serum ALP that predicted postoperative 30-day MACCE was 83 IU/L with a sensitivity and specificity of 76.9% and 72.1%, respectively (Figure 1).

## 4. Discussion

In this retrospective geriatric cohort study, we observed that preoperative serum ALP levels were associated with postoperative cardiac and cerebrovascular complications following lumbar spinal fusion surgery. The event rate for the composite endpoint of MI, stroke, and cardiovascular mortality within 30 days postoperatively after lumbar spinal fusion surgery was significantly higher in patients with the highest serum ALP tertile distribution (>79 IU/L) than in patients with lower values. Moreover, the highest serum ALP tertile was observed to be an independent predictor of MACCE, regardless of known cardiovascular risk factors.

The serum ALP level is primarily used as a biomarker of liver or metabolic bone disease. However, in recent times, it has been strongly implicated in vascular calcification and is associated with poor prognosis in patients with CKD and cardiovascular diseases, as well as in the general population [7,10,15]. Pathological vascular calcification is characterised by hydroxyapatite crystal growth, which is regulated by several factors, including pyrophosphate (a potent inhibitor of crystal propagation) [7]. Upregulation of tissue nonspecific ALP (TNALP) (comprising >90% of circulating ALP) secondary to uremia, injury, or genetic inheritance causes excessive hydrolysis and inactivation of pyrophosphate and promotes vascular calcification [8]. A previous in vitro study performed on bovine aortic smooth muscle cells that underwent oxidative stress-induced osteogenic differentiation reported overexpression of ALP in the smooth muscle cells [16]. A previous study reported increased expression and co-localisation of ALP with osteo/chondrogenic transcription factors in human arterial tissue samples [17]. Also, experimental genetic ablation or treatment with TNALP inhibitors ameliorated aortic calcification in rat models and accumulation of pyrophosphate in the cultured vascular smooth muscle cells [18]. Moreover, overexpression of ALP and enhanced mineralisation are known to occur as a cellular response to inflammatory stimuli [19,20], which explains the association between ALP activity and coronary or cerebral atherosclerotic disease that is primarily induced by immunoinflammatory stimulation [7,21].

Pathological tissue mineralisation primarily occurs in arteries and cardiac valves; therefore, elevated ALP activity can cause coronary and cerebrovascular calcification, arterial stiffness, and valvular heart disease [22]. Vascular aging is characterised by progressively enlarging deposits of calcium in the major arteries and cardiac valves. Thus, ALP activity could be associated with calcification of the aorta, the aortic valve, as well as the coronary and cerebral arteries in elderly patients. A previous in vitro study reported that prelamin A-induced premature senescence of vascular smooth muscle cells was associated with upregulation of messenger ribonucleic acid (RNA) and protein levels of ALP and osteogenic differentiation of cells [23]. A previous study with an 11-year follow-up has shown that in addition to its association with vascular aging itself, the serum ALP level in men aged 60–79 years without a history of MI or stroke served as a predictor of CAD and mortality [13]. In our study, elderly patients in the highest serum ALP tertile might have shown a greater degree of calcification and atherosclerosis of coronary and cerebral arteries and a consequently greater predisposition to perioperative vascular injury and subsequent ischemic events than patients of similar age with lower serum ALP levels. Moreover, perioperative hemodynamic might have been more severely compromised in patients with higher serum ALP levels, which concurs with a previous report that described reduced arterial elastance as an independent risk factor for blood pressure instability during the induction of general anesthesia in patients aged 60–80 years [24].

Increased TNALP expression or activity promotes endothelial dysfunction [7]. In a previously reported clinical trial that included 500 patients with essential hypertension, the authors observed that patients with higher serum ALP levels showed impaired endothelium-dependent vasodilatation [25], which plays an important role in maintaining haemodynamic stability and autoregulation of cardiac and cerebral circulation. Among patients with CAD undergoing drug-eluting stent implantation, those with higher serum ALP levels developed stent thrombosis or MI more frequently, which is attributable to the reduced number and function of endothelial progenitor cells [12]. Although aging itself causes an imbalance between reactive oxygen species and nitric oxide and consequent endothelial dysfunction [26], enhanced ALP activity accelerates endothelial cell senescence and dysfunction. This hypothesis is supported by previous experimental studies that report an association between TNALP expression and endothelial cell senescence in brain microvasculature resulting in blood-brain barrier breakdown [27,28]. Patients in the highest serum ALP tertile in our study may have shown a greater degree of pathological changes in the arterial endothelium, with consequent hemodynamic instability and atherothrombogenic events [29].

The predictive capacity of conventional risk factors for postoperative complications is reduced in elderly patients [30,31]. Therefore, it is important to identify reliable preoperative screening tools for cardiovascular risk stratification. Among the previously suggested tests, exercise electrocardiogram or stress echocardiography is the most reliable diagnostic modality [32]. However, it cannot always be performed in all types of surgical procedures that elderly patients undergo, such as ophthalmic, orthopedic, spine, or surgeries for a few cancers. Moreover, these procedures require the services of experienced technicians and may cause discomfort and/or harm to patients [33]. A coronary computed tomographic angiogram was shown to be useful to predict perioperative MI or mortality in patients with diabetes undergoing transfemoral amputation [34]; however, it may not be suitable for routine screening, considering the risk of adverse reactions or renal injury secondary to contrast dye, as well as for economic reasons. Notably, estimation of serum ALP levels is a cost-effective routine preoperative laboratory test that does not need any special equipment or expertise. To our knowledge, this is the first report that describes the utility of serum ALP as a predictor of postoperative cardiovascular complications after non-cardiac surgery in elderly patients. The optimal cut-off value of serum ALP that showed the best discriminatory capacity to predict the risk of postoperative MACCE in this study was 83 IU/L, which was close to the boundary of the highest serum ALP tertile that showed independent predictability (≥80 IU/L). This value is also consistent with the lower limits of ranges that predict poor prognosis in patients with MI or stroke [11,35]. The actual cut-off values of serum ALP that show significant prognostic power may differ across studies based on demographic factors and types of surgeries performed in patients. However, serum ALP levels > 80 IU/L in elderly patients without concomitant conditions known to cause increased ALP levels can be considered a risk factor for cardiovascular calcification and consequent perioperative cardiovascular morbidity.

The following are the limitations of our study: (1) The incidence of MACCE was extremely low (0.9%) for sufficient statistical verification, which is attributable to the fact that spinal surgery is currently a low-risk surgery. Lack of significant predictability of serum ALP levels for each endpoint can be understood in the same context. Homogeneity in the surgical procedure in the study cohort could ensure that our results were adequately reliable. However, further studies that include patients undergoing high-risk surgeries would be required to gain a better understanding of this subject. (2) We cannot completely exclude the possibility of confounders based on differences in the percentages of patients using beta-blockers among the study groups, although a history of beta-blocker use was not independently associated with the risk of MACCE. Data analysis after adjusting for possible confounders, such as comorbidities and beta-blocker use could have improved the reliability of our study. However, unfortunately this was not possible owing to a relatively small sample size. (3) The role of residual confounders secondary to systemic inflammation cannot be excluded [7]. However, to date, there is lack of confirmatory evidence showing a positive association between serum ALP levels and systemic inflammation. Previous reports describing the cardiovascular prognostic power of serum ALP levels, which is independent of C-reactive protein [10,12,13], support our argument that focuses on cardiovascular calcification as a mechanism that explains our results.

In conclusion, we observed that in patients aged >65 years undergoing lumbar spinal fusion surgery, higher preoperative serum ALP levels were independently associated with the composite endpoint of postoperative 30-day MI, stroke, or cardiovascular mortality. Our finding might implicate utility of preoperative serum ALP level as a potential biomarker of vascular calcification that plays a key role in causing cardiovascular complications in elderly patients. Further prospective studies to clarify the causal link between the serum ALP level and postoperative prognosis and verification on the other type of geriatric surgery are required.

## Figures and Tables

**Figure 1 jcm-08-01111-f001:**
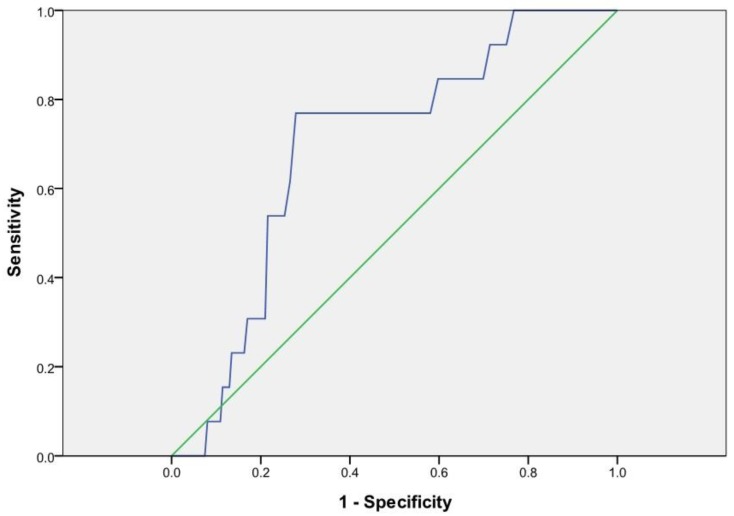
Combined receiver-operating characteristic curve of preoperative serum ALP levels for incidence of postoperative 30-day major adverse cardiac and cerebrovascular (MACCE). The area under the curve = 0.694 and *p*-value = 0.016 are observed below the line showing the serum ALP level with a 95% confidence interval of 0.574–0.813. The optimal cut-off value of serum ALP that predicted the incidence of MACCE was 83 IU/L, with a sensitivity and specificity of 76.9% and 72.1%, respectively. ALP; alkaline phosphatase, MACCE; major adverse cardiac and cerebrovascular events.

**Table 1 jcm-08-01111-t001:** Intergroup comparison of patient characteristics based on distribution of preoperative serum alkaline phosphatase levels into tertiles. Values are mean (SD), median (IQR (range)), or numbers (proportion).

ALP (IU/L)	Total	ALP < 63	ALP = 63–79	ALP > 79	*p*-Value
*n*	1395	468	463	464	
Age, yrs	72 ± 4	72 ± 4	72 ± 4	72 ± 5	0.758
Female	903 (64.7)	297 (63.5)	298 (64.5)	308 (66.4)	0.634
BMI, kg/m^2^	24.92 ± 3.01	24.82 ± 2.88	24.91 ± 3.02	25.04 ± 3.12	0.526
ASA class ≥ 3	453 (32.5)	148 (33.5)	144 (33.2)	161 (33.3)	0.450
Current smoker	109 (7.8)	39 (8.3)	38 (8.2)	32 (6.9)	0.665
Hypertension	822 (58.8)	275 (58.8)	270 (58.3)	277 (59.7)	0.909
Diabetes	304 (21.8)	104 (22.2)	94 (20.3)	106 (22.8)	0.620
Coronary artery disease	73 (5.2)	17 (3.6)	27 (5.8)	29 (6.3)	0.155
Chronic kidney disease	47 (3.4)	17 (3.6)	20 (4.3)	10 (2.2)	0.175
COPD	9 (0.6)	2 (0.4)	3 (0.6)	4 (0.9)	0.709
Preoperative Medication					
Beta blockers	139 (10.0)	43 (9.2)	60 (13.0)	36 (7.8)	0.024
CCBs	408 (29.2)	146 (31.2)	124 (26.8)	138 (29.7)	0.321
RAS blockers	521 (37.3)	173 (37.0)	172 (37.1)	176 (37.9)	0.949
Statins	370 (26.5)	123 (26.3)	124 (26.8)	123 (26.5)	0.981
Diuretics	191 (13.7)	57 (12.2)	65 (14.0)	69 (14.9)	0.473
LVEF, %	66.3 ± 6.2	66.5 ± 6.1	66.3 ± 6.0	66.0 ± 6.5	0.626
Laboratory Data					
Albumin, g/m	4.42 ± 0.46	4.45 ± 0.52	4.39 ± 0.35	4.42 ± 0.48	0.164
ALT, U/L	20 (16–27)	20 (16–26)	20 (15–27)	20 (16–29)	0.119
AST, U/L	23 (20–27)	23 (20–26)	23 (20–27)	24 (20–29)	0.087
Calcium, mg/dL	9.23 ± 0.89	9.27 ± 1.00	9.18 ± 0.64	9.26 ± 0.98	0.260
Total cholesterol, mg/dL	195.13 ± 45.57	196.12 ± 45.29	193.98 ± 42.80	195.28 ± 48.50	0.774
Creatinine, mg/dL	0.73 ± 0.23	0.74 ± 0.25	0.72 ± 0.21	0.73 ± 0.22	0.384
Hemoglobin, g/dL	13.38 ± 1.31	13.31 ± 1.29	13.44 ± 1.23	13.39 ± 1.42	0.360
Total bilirubin, mg/dL	0.61 ± 0.25	0.60 ± 0.24	0.60 ± 0.25	0.63 ± 0.26	0.130
Phosphorus, mg/dL	3.76 ± 0.60	3.77 ± 0.60	3.77 ± 0.55	3.72 ± 0.65	0.381
ALP, IU/L	74.27 ± 24.93	52.04 ± 7.29	70.73 ± 4.91	100.23 ± 24.61	< 0.001

Values are mean (SD), median (IQR (range)), or numbers (proportion). SD; standard deviation, IQR; interquartile range, ALP; alkaline phosphatase, BMI; body mass index, ASA class; American Society of Anesthesiologists classification, COPD; chronic obstructive pulmonary disease, CCBs; calcium-channel blockers, RAS; renin-angiotensin system, LVEF; left ventricular ejection fraction, ALT; alanine transaminase, AST; aspartate transaminase.

**Table 2 jcm-08-01111-t002:** Intergroup comparison of perioperative data based on distribution of preoperative serum alkaline phosphatase levels into tertiles.

ALP (IU/L)	Total	ALP < 63	ALP = 63–79	ALP > 79	*p*-Value
*n*	1395	468	463	464	
Intraoperative Data					
Number of fusion level	1 (1–2)	2 (1–2)	1 (1–2)	1 (1–2)	0.233
Multi-level fusion (>3)	74 (5.3)	29 (6.2)	16 (3.5)	29 (6.3)	0.095
Surgery time, min	230.61 ± 74.11	234.62 ± 72.77	227.34 ± 74.19	229.83 ± 75.33	0.313
Fluid, mL	2200 (1650–2850)	2200 (1700–2850)	2150 (1650–2750)	2150 (1650–2900)	0.630
Bleeding, mL	700 (500–1100)	700 (500–1100)	650 (500–1100)	800 (500–1200)	0.114
Number of patients transfused with pRBCs	343 (24.6)	114 (24.4)	104 (22.5)	125 (26.9)	0.283
RBC transfused, mL	0 (0–0)	0 (0–0)	0 (0–0)	0 (0–240)	0.317
Urine Output, mL	600 (350–1000)	550 (330–950)	600 (350–1000)	635 (350–1000)	0.329
Postoperative Data					
Myocardial infarction	9 (0.6)	2 (0.4)	1 (0.2)	6 (1.3)	0.095
Stroke	5 (0.4)	1 (0.2)	0 (0.0)	4 (0.9)	0.073
Death	1 (0.1)	0 (0.0)	0 (0.0)	1 (0.2)	0.366
Composite (MACCE)	13 (0.9)	2 (0.4)	1 (0.2)	10 (2.2)	0.003
MV > 24 h	4 (0.3)	2 (0.4)	1 (0.2)	1 (0.2)	0.784
ICU admission	73 (5.2)	24 (5.1)	25 (5.4)	24 (5.2)	0.980
LOS after surgery, day	10 (9–12)	10 (9–12)	10 (9–12)	11 (9–13)	0.097

Values are mean (SD), median (IQR (range)), or numbers (proportion). SD; standard deviation, IQR; interquartile range, ALP; alkaline phosphatase, pRBCs; packed red blood cells, RBC; red blood cells, MACCE; major adverse cardiac and cerebrovascular events, MV; mechanical ventilation, ICU; intensive care unit, LOS; length of stay.

**Table 3 jcm-08-01111-t003:** Intergroup comparison of baseline characteristics and operative data based on the incidence of 30-day major adverse cardiac and cerebrovascular events.

	No MACCE	MACCE	*p*-Value
*n*	1382	13	
Age, years	72 ± 4	72 ± 4	0.683
Female sex	892 (64.5)	11 (84.6)	0.132
BMI, kg/m^2^	24.92 ± 3.01	24.76 ± 2.72	0.921
ASA class ≥ 3	450 (32.6)	3 (23.1)	0.467
Hypertension	814 (58.9)	8 (61.5)	0.847
Diabetes	299 (21.6)	5 (38.5)	0.144
Coronary artery disease	72 (5.2)	1 (7.7)	0.690
Chronic kidney disease	46 (3.3)	1 (7.7)	0.385
COPD	9 (0.7)	0 (0.0)	0.770
Preoperative Medication			
Beta blockers	138 (10.0)	1 (7.7)	<0.999
CCBs	403 (29.2)	5 (38.5)	0.541
RAS blockers	516 (37.3)	5 (38.5)	<0.999
Statins	365 (26.4)	5 (38.5)	0.348
Diuretics	190 (13.7)	1 (7.7)	0.451
Preoperative Laboratory Data			
Hemoglobin	13.38 ± 1.31	13.18 ± 1.11	0.342
Creatinine	0.73 ± 0.23	0.80 ± 0.27	0.464
ALP, IU/L	74.2 ± 24.9	84.5 ± 15.4	0.139
First/Second/Third tertiles	466 (33.7)/462 (33.4)/454 (32.9)	2 (15.3)/1 (7.6)/10 (76.9)	0.006
Multi-level fusion (> 3)	73 (5.3)	1 (7.6)	0.509
Operation time, min	221 (181–270)	225 (165–315)	0.438
Bleeding, mL	700 (500–1100)	800 (550–1100)	0.571
RBC transfused, mL	0 (0–0)	0 (0–360)	0.541

Values are mean (SD), median (IQR (range)), or number (proportion). SD; standard deviation, IQR; interquartile range, MACCE; major adverse cardiac and cerebrovascular events, BMI; body mass index, ASA class; American Society of Anesthesiologists classification, COPD; chronic obstructive pulmonary disease, CCBs; calcium channel blockers, RAS; renin-angiotensin system, ALP; alkaline phosphatase, RBC; red blood cells.

**Table 4 jcm-08-01111-t004:** Logistic regression analysis of variables predicting incidence of 30-day major adverse cardiac and cerebrovascular events.

	Univariate		Multivariate	
	OR (95% CI)	*p*-Value	OR (95% CI)	*p*-Value
Age, years	0.993 (0.874–1.127)	0.910		
Female sex	3.021 (0.667–3.021)	0.151	2.913 (0.641–13.243)	0.166
Diabetes	2.264 (0.735–6.971)	0.154	2.188 (0.707–6.778)	0.175
Coronary artery disease	1.515 (0.194–11.813)	0.692		
Chronic kidney disease	2.420 (0.308–19.010)	0.401		
ALP third tertile	4.599 (1.409–15.014)	0.011	4.507(1.378–14.739)	0.013
Bleeding amount, mL	1.002 (1.000–1.004)	0.387		

Values are mean (SD), median (IQR (range)), or number (proportion). SD; standard deviation, IQR; interquartile range, OR; odds ratio, CI; confidence interval, ALP; alkaline phosphatase.

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
