# Peer review of "Serum Alkaline Phosphatase as a Predictor of Cardiac and Cerebrovascular Complications after Lumbar Spinal Fusion Surgery in Elderly: A Retrospective Study"

_jcm, 2019, doi:10.3390/jcm8081111_

Reviewer 1 Report

Interesting paper investigating the significance of ALP values.

 The study has some limits already indicated by Authors.

Since all data were collected and analyzed in patients undergoing spinal fusion surgery, I would suggest or to validate results and hypothesis in patients with at least an other type of surgery or to specifically limit the study of one type of patients modifying the title that, at present, gives the impression of a more generalized trend. 

Author Response

 As you pointed out, presenting as “non-cardiac surgery” may exaggerate or generalize the current result for all the geriatric surgical patients. We have specified the patient cohort as those undergoing lumbar spinal fusion surgery on the title as well as the abstract and manuscript.

Reviewer 2 Report

An overall very good presentation of an equally designed study. Detailed information about methodology, analysis and clear mention of the limitations. Rich relative information about the role of ALP with a lot of updated references.

Given the limitations and the conclusion that authors mentioned, could they provide a practical "conclusion- guide" to when and when not and how to use ALP as possible marker for post operative complications?It would be an extremely helpfull suggestion for the daily practice clinician.

Yet, if they fill that we are not ready to confirm the use of ALP for such purpose- and the present results are only a possitive "clue"; let , please , give some future road paths of reseach more detailly.

The mention some in the limitations section, yet can they plan the next steps?

This would be helpful to all colleagues for future research.

Author Response

We appreciate your suggestion to provide more conclusive and detail information regarding using the serum ALP level as a preoperative assessment. However, as you mentioned, several defects of this study, such as small sample size and limited surgical type, may restrict any solid conclusion of the clinical implication of ALP level in the perioperative outcomes in elderly. Nevertheless, we are pleased to provide a primary evidence regarding its utility as a possible preoperative risk factor in this old age population. Next step should be extending and/or including the high risk surgeries with larger sample size to verify our result. Also, mechanistic insights should be provided by well-controlled prospective trial. We have added the paragraph suggesting future research needed to the Conclusion section.